# Critically Important Antimicrobial Resistance Trends in *Salmonella* Derby and *Salmonella* Typhimurium Isolated from the Pork Production Chain in Brazil: A 16-Year Period

**DOI:** 10.3390/pathogens11080905

**Published:** 2022-08-11

**Authors:** Caroline Pissetti, Eduardo de Freitas Costa, Karoline Silva Zenato, Marisa Ribeiro de Itapema Cardoso

**Affiliations:** 1Department of Preventive Veterinary Medicine, Faculty of Veterinary, Federal University of Rio Grande do Sul, Porto Alegre 91540-000, RS, Brazil; 2Department of Epidemiology, Bio-Informatics and Animal Models, Wageningen Bioveterinary Research, 8221 RA Lelystad, The Netherlands

**Keywords:** antimicrobial resistance, carbapenems, colistin, ESBL, ciprofloxacin

## Abstract

Knowledge about antimicrobial resistance in *Salmonella* is relevant due to its importance in foodborne diseases. We gathered data obtained over 16 years in the southern Brazilian swine production chain to evaluate the temporal evolution of halo for carbapenem, and the MIC for third-generation cephalosporins, fluoroquinolone, and polymyxin in 278 *Salmonella* Derby and Typhimurium isolates. All antimicrobial resistance assays were performed in accordance with EUCAST. To assess the diameter halo, we used a mixed linear model, and to assess the MIC, an accelerated failure time model for interval-censored data using an exponential distribution was used. The linear predictor of the models comprised fixed effects for matrix, serovar, and the interaction between year, serovar, and matrix. The observed halo diameter has decreased for ertapenem, regardless of serovars and matrices, and for the serovar Typhimurium it has decreased for three carbapenems. The MIC for ciprofloxacin and cefotaxime increased over 16 years for Typhimurium, and for Derby (food) it decreased. We did not find evidence that the MIC for colistin, ceftazidime, ciprofloxacin (Derby), or cefotaxime (food Typhimurium and animal Derby) has changed over time. This work gave an overview of antimicrobial resistance evolution from an epidemiological point of view and observed that using this approach can increase the sensitivity and timeliness of antimicrobial resistance surveillance.

## 1. Introduction

Bacterial resistance to antimicrobial agents is a growing global problem, stemming from the selection that occurs through the use of antimicrobials in humans, animals, and the environment. Veterinary-prescribed antimicrobials for prophylactic, therapeutic, or growth promotion purposes are often the same or closely related to those used in human medicine [1,2]. Therefore, bacteria with antimicrobial resistance profile present in farm animals may pose a risk to animal and public health [3]. These bacteria can be transmitted to humans directly or via food [4], causing infections often with limited treatment due to their multidrug resistance profile [5]. According to Antimicrobial Resistance Collaborators, in a systematic review, [6] there were an estimated 4.95 million deaths associated with bacterial antimicrobial resistance in 2019, including 1.27 million deaths attributable to bacterial antimicrobial resistance.

*Salmonella* is one of the most important foodborne agents related to animal production in several countries, including Brazil [7,8,9,10]. According to de Freitas Costa et al. [11], *Salmonella* is the most relevant hazard to public health in the Brazilian pork production chain. The prevalence of pre-chilling carcasses in Brazilian slaughterhouses ranges from 10.2 to 35.7% [12,13,14,15], and Derby and Typhimurium have been the most frequent serovars isolated from pigs and pork in Brazil [12,13,14,16,17,18]. During many years, both serovars have been extensively isolated and studied in this target population, but investigations on resistance to antimicrobials on the list of the highest priority critically important and carbapenem in isolates of *Salmonella* from the swine production chain are still limited in Brazil [15,16,18,19].

The monitoring of antimicrobial agents in animal production first took place in 1995 in Denmark [20]. Since 2003, European Union member states report antimicrobial resistance data of *Salmonella* on a mandatory basis, allowing researchers to report temporal trends on *Salmonella* resistance in the long term [21]. In the United States of America, since 1996 there is in place the National Antimicrobial Resistance Monitoring System for Enteric Bacteria (NARMS), and its data have been used to track trends in antimicrobial resistance in humans, retail meat, and food animals [22,23,24].

In 2019, the Brazilian Ministry of Agriculture, Livestock, and Food Supply (MAPA initials in Portuguese) implemented the national Monitoring and Surveillance Program for Antimicrobial Resistance in Agriculture. The program focuses on *Salmonella* spp. in the farm (feces) and slaughterhouse (intestinal content and carcasses) of swine [25], but it is still in a phase of data collection and, consequently, up to now, little is known about the temporal patterns for antimicrobial resistance in Brazil. Most of the knowledge about *Salmonella* in the Brazilian pig production chain was generated in different research projects, aiming to assess the factors associated with the presence of *Salmonella* in pre/post-harvest. The interest in antimicrobial resistance has increased over time, Rodrigues et al. [17] conducted a systematic review in Brazil that includes *Salmonella* isolates of swine-origin for three decades, but with the objective of descriptive analysis. Although the description over time gives some picture of the antimicrobial resistance, it does not have the robustness to test hypotheses, or allows making statistical inferences about the evaluation of the studied phenomenon trends over time.

To provide inferences on the temporal evolution of the antimicrobial resistance in pork production in Brazil before the implementation of the national Monitoring and Surveillance Program for Antimicrobial Resistance in Agriculture, we gathered secondary data from research projects obtained during 16 years from the swine production chain of southern Brazil. We assessed the temporal evolution of the halo diameter to carbapenem (ertapenem, imipenem, and meropenem), and the minimum inhibitory concentration to third-generation cephalosporins (cefotaxime and ceftazidime), fluoroquinolone (ciprofloxacin), and polymyxin (colistin) in strains of *Salmonella* Derby and Typhimurium.

## 2. Results

### 2.1. Antimicrobial Resistance Trends

The analysis of the resistance profile of *Salmonella* Typhimurium and Derby grouped according to their origin (animal or food) showed different trends over time for the critically important antimicrobials tested.

#### 2.1.1. Carbapenems

All isolates showed susceptibility to the tested carbapenems, according to the clinical cut-off point established by EUCAST [26]. When the epidemiological cut-off (ECOFF) value was considered, which was available only for meropenem (27 mm) [27], all strains were classified as wild type. Considering the screening cut-off values for carbapenemase-producing *Enterobacterales*, according to EUCAST methodology [28,29] no strain was positive or needed to be submitted to additional tests.

However, several strains classified as susceptible and wild type showed a trend of a decreasing inhibition halo diameter in the disk diffusion test over time. This trend could be observed in the serovar Typhimurium regardless of the antibiotic tested and origin of the strain. The same trend was observed in serovar the Derby only for ertapenem, while for imipenem and meropenem, stability on the inhibition halo diameter was observed (Figure 1).

In this sense, ertapenem stood out as the antimicrobial of the carbapenem group in which the reduction trend could consistently be evidenced. The halo diameters in millimeters (mm) for all samples, regardless of serovar and origin, showed a reducing trend over time, and the slope coefficient for the interaction terms showed the most accentuated reduction for strains originated from animals. For each year, the mean halo diameter reduced by 0.36 and 0.31 mm for the serovars Derby and Typhimurium, respectively (Table 1). When meropenem was considered, the reduction trend was significant only for the serovar Typhimurium: a reduction of 0.29 mm per year, on average, in samples from animals (Table 1). The halo diameter reduction trend when imipenem was considered was significant only for Typhimurium strains from animals, which showed a reduction of 0.24 mm per year, on average (Table 1). In all other combinations, we could not reject the null hypothesis, and based on this, it is not possible to infer that the halo diameter has changed over time. The intracluster correlation for all models was lower than 20%, and a moderate proportion of the variation in the halo diameter was explained by the study in which the strains were isolated.

#### 2.1.2. Third-Generation Cephalosporins, Ciprofloxacin and Colistin

Regarding the MIC results found for third-generation cephalosporins, only one Typhimurium strain of animal origin showed resistance according to the breakpoint established by EUCAST [26] for ceftazidime (Table 2). Considering the screening breakpoint of > 1 mg/L for extended-spectrum β-lactamase (ESBL)-producing *Enterobacterales*, 13 strains were submitted to the double-disk synergy test (DDST), but none confirmed the presence of ESBL. When considering ECOFF values, a total of 14 strains were classified as non-wild type: eleven for ceftazidime, one for cefotaxime, and two for both antimicrobials.

Regarding ciprofloxacin, just the serovar Typhimurium showed resistance, totaling 77 strains, which can be observed in Figure 2. Moreover, 117 isolates were classified as not being wild type. The highest number of resistant strains was observed against colistin, with 130 strains being resistant. Again, the most expressive number of resistant strains were found in the serovar Typhimurium from animal origin. None of the resistant strains carried the *mcr-*1 gene. One colistin-resistant strain also showed resistance to ceftazidime, and 15 colistin-resistant strains were also resistant to ciprofloxacin.

In Typhimurium strains originated from animals and food, it was estimated that there was an average increase in MIC for ciprofloxacin by 28 and 26% per year (i.e., $e^{\beta }-1$), respectively (Table 3). For cefotaxime, Typhimurium strains from animals showed a MIC value increase by 3% per year, on average. On the other hand, strains from the serovar Derby from food had a decreasing MIC by 5.8% per year, on average (Table 3). The MIC values for colistin and ceftazidime presented no change over time estimated by the model.

## 3. Discussion

We showed temporal trends for the halo diameter and MIC values for *Salmonella* Typhimurium and Derby strains isolated from the Brazilian pig production chain from 2000 to 2015. To the best of the authors’ knowledge, this is the first study attempting to explore the hypothesis on the temporal trends for antimicrobial resistance in Brazil. So far, most of the Brazilian studies aimed to describe the prevalence and genetic diversity of *Salmonella* strains isolated from the pig production chain at a given time point [15,18,30,31,32]. More recently, Rodriguez et al. [17] performed a systematic review to assess the antimicrobial resistance in *Salmonella* isolated from swine between 1990 and 2016 in Brazil. In this study, the authors explored the evolution over time in a descriptive way. Here we included the time as a covariate in generalized linear models, allowing for the assessment of possible signaling of increasing on epidemiological resistance over time. This approach may be useful for early-warning surveillance on antimicrobial resistance [33,34].

It is possible to observe a consistency in the results for the serovar Typhimurium, regarding the decreasing and increasing the halo diameter and MIC, respectively. Serovar Typhimurium has been reported as being prone to acquire resistance in comparison with other serovars, including Derby [15,17,18]. This tendency was also confirmed in the analyzed strains from this study. In addition, it is possible to observe that strains from animal samples (i.e., intestinal content) tended to increase their resistance over time. A possible explanation for this may be the fact that the animal intestine is a site for bacterial resistance selection, related to the frequent antimicrobial administration via feed.

In this study, we detected a high number of strains displaying clinical resistance to colistin. Since 2016, colistin is allowed in Brazil only for treatment of diseased pigs [35]; however, before the ban of its use as a growth promoter, colistin had been used intensively. This fact may have contributed for the high level of resistance observed among the *Salmonella* strains and for the stability of high MIC values over time in Typhimurium strains from pigs. Once considered as resistance related to a mutation in the bacterial chromosome, the discovery of the *mcr*-1 plasmidial gene in *E. coli* shed light on a more spreadable resistance mechanism [36]. After the first report, *mcr*-1 has been found in species of *Enterobacterales* isolated in several countries, including in strains isolated before being first observed [37]. Interestingly, the presence of *mcr*-1 was not detected in our strains, indicating that the resistance may be chromosomally encoded or related to other less frequent genes carried on plasmids, including others *mcr* genes [38,39].

We observed an increase in MIC to ciprofloxacin over the 16 years in the serovar Typhimurium. The reported resistance of *Salmonella* against ciprofloxacin in Brazilian swine production is high [16,31,40]. In Brazilian pig farms, both enrofloxacin and ciprofloxacin are used for therapeutic and prophylactic purposes [41]. This may have been the driver for resistance selection, resulting in the observed increase in MIC overtime. From a surveillance perspective, although the observed MICs are lower than the clinical cut-off for resistance, their increase should be a warning for risk managers about a possible deterioration of the situation concerning an important antimicrobial drug for animals and humans.

Among the isolates tested against third-generation cephalosporins, clinical resistance was observed, increasing by 3% per year in cefotaxime MIC values for Typhimurium strains of animal origin. Therapeutically, cephalosporin is often used in swine production, especially ceftiofur [42]. However, when compared to other antimicrobials, the use of this class is less frequent, and its administration is exclusively intramuscular. Even so, the selection of resistant isolates can occur [43]. In addition, ESBL-producing strains were not detected by the phenotypic tests, reinforcing that the presence of resistant strains is still rare in *Salmonella* isolated from swine in Brazil [18,44].

Regarding carbapenems, we observed a consistent reduction in zone diameter for ertapenem, imipenem and meropenem over the years in Typhimurium isolated from animal samples, alerting to a possible ongoing resistance selection. Despite the carbapenem antimicrobials being not approved or used in farm animals, co-selection of carbapenem resistance by other veterinary drug classes (e.g., β-lactams, fluoroquinolones, cephalosporins, tetracyclines) has been documented [45,46] and may be an additional concern. A possible limitation is that we based the observation not on MIC results but on the inhibition diameter zone of disk diffusion tests. However, this latter parameter has been used by EUCAST to monitor the epidemiological cut-off point, considering that the disk diffusion test is highly used in routine diagnostic tests.

The strains analyzed in this study were obtained from several studies conducted mainly in the south region of Brazil. Although it was not a designed sampling strategy for a national representation, historically, this region concentrates approximately 50% of Brazilian pork production [47]. This gives a good picture of antimicrobial resistance in the most important and dense area regarding pork production in the country. Another consequence of using secondary data for temporal trending is the possible sparsity of data in a few years. For food samples, there are fewer data in 2002, and for animal data, there are fewer data after 2012. The lower frequency of observations in the time domain does not invalidate the analysis, however, it penalizes the standard error [48] jeopardizing the power to detect significant effects (i.e., temporal trends) [49].

Even with limitations, this study gives a picture of the evolution of antimicrobial resistance from an epidemiological standpoint, covering a period of 16 years before the Brazilian antimicrobial resistance control program. The results observed here contribute to depicting the evolving situation of *Salmonella* resistance before the implemented program in Brazil, helping to shape surveillance and to direct efforts; meanwhile, the newly implemented program is still gathering data. Further studies could overcome the cited limitations, gathering more historical data to increase the power of the analyses. Including data after the implementation of the control program could give a comparative picture of the effect of measures adopted in the pig sector on the antimicrobial resistance trends. Also, the approach used may increase the sensitivity and timeliness of antimicrobial resistance monitoring. Finally, the results about the evolving situation for carbapenem and ciprofloxacin may be useful for monitoring the future evolution of these critically important antimicrobials for human.

## 4. Materials and Methods

### 4.1. Bacterial Isolates

In total 278 *Salmonella* Typhimurium isolates and 135 *Salmonella* Derby isolates from the culture collection of the Preventive Veterinary Medicine Department—Faculty of Veterinary Medicine of the Federal University of Rio Grande do Sul (FAVET/UFRGS) were used in this study. The isolates come from about 3000 samples collected from 2000 to 2015, from 11 different master’s and doctoral projects (Table 4), in swine slaughterhouses located in the states of Santa Catarina and Rio Grande do Sul, where about 50% of Brazilian swine production is concentrated [47]. The strains were divided into two categories: *i.* animal—corresponding to intestinal content and lymph nodes; *ii.* food—corresponding to carcasses, and products of swine origin. Isolates from different matrices of the same animal were not used in the study.

### 4.2. Antimicrobial Resistance Test

The same person under the same laboratory conditions and in accordance with European Committee on Antimicrobial Susceptibility Testing (EUCAST) guidelines performed all antimicrobial resistance assays.

A screening test was performed to detect carbapenemase-producing microbes by the disc diffusion method [28,29]. For this, the screening cut-off to ertapenem (10 µg, Oxoid) and meropenem (10 µg, Oxoid) was an inhibition halo of <25 mm [29] and imipenem (10 µg, Oxoid) was <23 mm. *Klebsiella pneumoniae* NCTC 13442 was used for the control test. Inhibition halo diameters were evaluated for their medians and distributed into quantiles for each antimicrobial and serovar. The diameter values were compared with breakpoints and epidemiological cut-offs (ECOFF) for meropenem [27].

Minimum inhibitory concentration (MIC) was determined by the broth microdilution method for the following antimicrobials: cefotaxime, ceftazidime, ciprofloxacin (Sigma Aldrich, St. Louis, MO, USA), and colistin (European Pharmacopoeia, United Kingdom). MIC values were interpreted according to clinical breakpoints and ECOFF available by EUCAST [26]. The *Escherichia coli* ATCC 25922 strain was used as a control test. Strains showing MICs > 1 mg/L for cefotaxime or ceftazidime were submitted to a phenotypic confirmatory test for extended-spectrum β-lactamase (ESBL) by double-disk synergy test (DDST) using cefotaxime (5 µg, Oxoid, Basingstoke, UK) and ceftazidime (10 µg, Oxoid, Basingstoke, UK) in approximation with same antimicrobial discs associated with clavulanic acid (Oxoid, Basingstoke, UK) [29]. *Klebsiella pneumoniae* ATCC 700603 and *E. coli* ATCC 25922 were used, respectively, as positive and negative controls for the control test. The strains with MICs ≥ 2 mg/L for colistin were submitted to *mcr*-1 gene analysis by PCR [36] using an *Escherichia coli*, previously studied and confirmed as positive for *mcr*-1 gene, as a positive control [44].

### 4.3. Statistical Analysis

To assess the halo diameter (y), we used a linear mixed model using the identity link function. The linear predictor of the model comprised fixed effects for serovar *x*1 {Typhimurium = 0 or Derby = 1}, matrix *x*2 {animal = 0 or food = 1}, and the interaction between time *x*3 {2000,...,2015}, serovar and matrix. The project in which the sample was taken was used as a random-intercept [58]:(1)$$y_{i,j}=\beta_{0}+\beta_{Derby}\cdot x1_{i,j}+\beta_{food}\cdot x2_{i,j}+\beta_{time:serovar:matrix}\cdot {\left(x1\cdot x2\cdot x3\right)}_{i,j}+u_{j}+e_{i,j},$$
where $\beta_{0}$ is the intercept for serovar Typhimurium and matrix animal, and $\beta_{Derby}$, $\beta_{food}$, are the changes in $\beta_{0}$ for serovar Derby and matrix food, respectively. $\beta_{time:serovar:matrix}$, is a vector of coefficients for the interaction of time, matrix and serovar, representing the linear change in the halo diameter given the increase in the year (slope).

The error term for residuals $e_{i,j}\sim N\left(0,\sigma_{e}\right)$ and the random project effect $u_{j}\sim N\left(0,\sigma_{u}\right)$ were assumed be normally and independently distributed. Consequently, the intra cluster correlation (ICC) equals $\sigma_{u}/\left(\sigma_{u}+\sigma_{e}\right)$.

The hypothesis of temporal trend was tested according to:(2)$$\left\{\begin{array}{c} \beta_{time:serovar:matrix}=0\\ \beta_{time:serovar:matrix}\neq \;0 \end{array}\right.,$$
notice that the null hypothesis states that there is no trend in the halo diameter over time. The null hypothesis for the intercepts coefficients means that there is no deviation from intercept for the serovar Typhimurium and matrix animal ($\beta_{0}$), from the intercepts for serovar Derby $\left(\beta_{Derby}\right)$ and food $\left(\beta_{food}\right)$.

For assessing the MIC, an accelerated failure time (AFT) model survival regression for interval-censored data was used assuming an exponential distribution for the MIC (i.e., exponential link function). The “time” referred to in the AFT analysis can be extrapolated to any positive continuous asymmetric distribution such as MIC. According to Björk et al. [59], AFT regression methods minimize information loss, compared to techniques that categorize the MIC values. Also, it enables one to accommodate interval-censored data. Thus, AFT models detect effects missed in other regression models, such as logistic regression, making them a useful tool for surveillance. The linear predictor of the model comprised fixed effects for matrix (food or animals), serovar (Typhimurium or Derby), and the interaction between year, serovar and matrix. The project in which the sample was taken was used as a cluster, used in computing the robust variance [60]. The linear equation and the hypothesis of temporal trend followed the same structure for the halo diameter (Equations (1) and (2)).

More details about the accelerate failure mode are available in Appendix A. All analysis were conducted in R [61]. For halo diameter we used the routine lmer from library lmer4 [58]. For MIC, we used the routine survreg from library survival [62,63]. The data and scripts are available on the public repository: https://github.com/eduardodefreitascosta/AMR_surveillance. (accessed on 30 June 2022)

## Figures and Tables

**Figure 1 pathogens-11-00905-f001:**
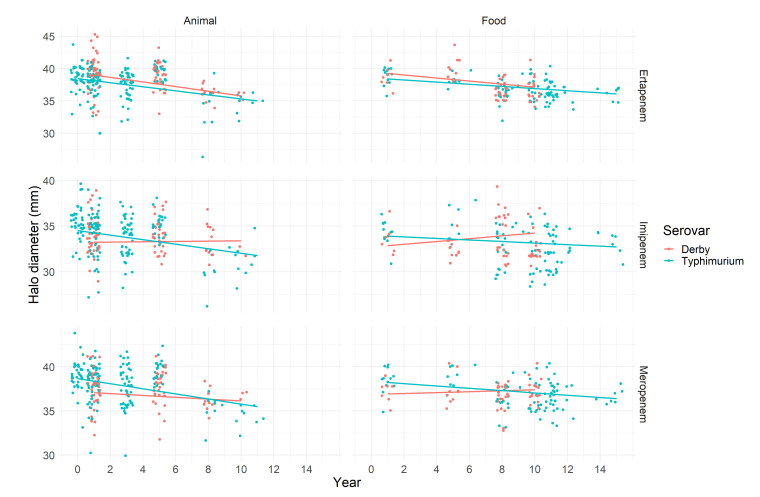
Observed halo diameter (dots) in millimeters for *Salmonella* Typhimurium and Derby isolated from animals and food samples between 2000 and 2015 tested against carbapenems (ertapenem, imipenem, and meropenem). Solid lines are the linear predicted trend.

**Figure 2 pathogens-11-00905-f002:**
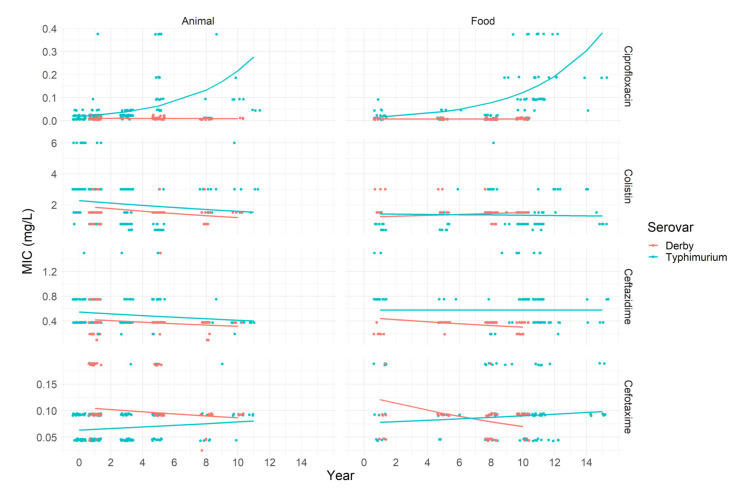
Observed median minimum inhibitory concentration (MIC) (dots) in mg/L for *Salmonella* Typhimurium and Derby isolated from animals and food samples between 2000 and 2015 tested against ciprofloxacin, colistin, ceftazidime, and cefotaxime. Solid lines are the predicted MIC.

**Table 1 pathogens-11-00905-t001:** Results for the linear mixed-effects model for assessing the halo diameter in millimeters (mm) against ertapenem, meropenem, and imipenem considering time, the serovars Typhimurium and Derby, and samples obtained from animals and food during the years 2000 and 2015.

Variable ^1^	Estimate	95% CI ^2^	*p*-Value	ICC ^3^
Ertapenem:				19.41%
$\beta_{0}$	39.36	38.16; 40.53	-	
$\beta_{Derby}$	0.91	0.17; 1.7	0.02	
$\beta_{food}$	0.13	−1.04; 1.36	0.83	
$\beta_{time,\;Derby,\;animal}$	−0.36	−0.57; −0.14	0.003	
$\beta_{time,\;Typhimurium,animal}$	−0.31	−0.48; −0.14	0.001	
$\beta_{time,\;Derby,\;food}$	−0.23	−0.43; −0.04	0.03	
$\beta_{time,\;Typhimurium,food}$	−0.16	−0.32; −0.01	0.04	
Imipenem:				16.8%
$\beta_{0}$	33.21	32.00; 34.39	-	
$\beta_{Derby}$	−1.3	−2.03; −0.47	0.001	
$\beta_{food}$	−0.50	−1.70; 0.66	0.41	
$\beta_{time,\;Derby,\;animal}$	0.02	−0.21; 0.23	0.87	
$\beta_{time,\;Typhimurium,\;animal}$	−0.24	−0.42; −0.08	0.01	
$\beta_{time,\;Derby,\;food}$	0.15	−0.05; 0.34	0.13	
$\beta_{time,\;Typhimurium,food}$	−0.08	−0.23; 0.06	0.28	
Meropenem:				15.20%
$\beta_{0}$	37.18	36.18; 38.17	-	
$\beta_{Derby}$	−1.48	−2.15; −0.77	<0.001	
$\beta_{food}$	−0.31	−1.35; 0.75	0.57	
$\beta_{time,\;Derby,\;animal}$	−0.10	−0.29; 0.08	0.29	
$\beta_{time,\;Typhimurium,\;animal}$	−0.29	−0.43; −0.14	<0.001	
$\beta_{time,\;Derby,\;food}$	0.05	−0.11; 0.22	0.54	
$\beta_{time,\;Typhimurium,food}$	−0.13	−0.26; −0.002	0.06	

^1^$\beta_{0}$ stands for the intercept for serovar Typhimurium and matrix food. $\beta_{Derby}$ is the increment in the $\beta_{0}$ for the serovar Derby, and $\beta_{food}$ is the increment in the $\beta_{0}$ for the matrix food.$\;\beta_{time}$ coefficients are the linear change in the halo diameter per increase of year (slope) for the different combinations of serovar and matrix. ^2^ CI = Confidence interval. ^3^ Variance components for Ertapenem model: ($\sigma_{u_{j}}=0.95;\sigma_{e}=3.9$); Variance components for Imipenem model: ($\sigma_{u_{j}}=0.81;\sigma_{e}=4$); Variance components for Meropenem model: ($\sigma_{u_{j}}=0.58;\sigma_{e}=3.2$).

**Table 2 pathogens-11-00905-t002:** Minimum Inhibitory Concentration (MIC) results of *Salmonella* Derby (*n* = 135) and *Salmonella* Typhimurium (*n* = 278) isolates against different antimicrobials.

ATM	Source	Serovar	Number of Isolates with MIC (mg/L) Same to:												ECOFF ^a^	R ^b^
			0.008	0.015	0.03	0.06	0.125	0.25	0.5	1	2	4	8	16	(%)	(%)
**Ciprofloxacin**	**Animal**	Derby	40	30	12										0	0
		Typhimurium	31	29	66	28	11	5	8	2					30	14.44
	**Food**	Derby	37	16											0	0
		Typhimurium	12	10	13	12	25	13	8	5 ^c^					64.29	52.04
**Colistin**	**Animal**	Derby								11	61	10			*	12.20
		Typhimurium							15	27	42	85	10	1	*	53.33
	**Food**	Derby								5	42	6			*	11.32
		Typhimurium							10	46	24	17	1		*	18.37
**Ceftazidime**	**Animal**	Derby					4	12	48	17	1				1.22	0
		Typhimurium						5	107	64	3	1			2.22	0.56
	**Food**	Derby						14	39						0	0
		Typhimurium						5	41	44	8				8.16	0
**Cefotaxime**	**Animal**	Derby			1	7	55	17	1	1					2.44	0
		Typhimurium				94	81	5							0	0
	**Food**	Derby				16	35	2							0	0
		Typhimurium				31	47	19	1						1.02	0

^a^ Epidemiological cutoff (ECOFF) values; ^b^ Resistant strains (R), as clinical breakpoint values; Gray areas represent MIC values above resistance breakpoint; | represent MIC values above ECOFF; ^c^ 4 strains showed MIC > 1; * No data available.

**Table 3 pathogens-11-00905-t003:** Results for the accelerated failure model assessing the minimum inhibitory concentration (MIC) (mg/L) against ciprofloxacin, colistin, ceftazidime, and cefotaxime considering time, the serovars Typhimurium and Derby, and samples obtained from animals and food during the years 2000 and 2015.

Variable ^1^	Estimate	95% CI ^2^	*p*-Value
Ciprofloxacin:			
$\beta_{0}$	−4	−4.41; −3.64	-
$\beta_{Derby}$	−0.56	−0.81; −0.31	<0.001
$\beta_{food}$	−0.36	−0.99; 0.26	0.27
$\beta_{time,\;Derby,\;animal}$	−0.02	−0.09; 0.05	0.62
$\beta_{time,\;Typhimurium,\;animal}$	0.25	0.06; 0.43	0.01
$\beta_{time,\;Derby,\;food}$	0	−0.07; 0.07	0.99
$\beta_{time,\;Typhimurium,food}$	0.23	0.16; 0.29	<0.001
Colistin:			
$\beta_{0}$	0.82	0.55; 1.09	-
$\beta_{Derby}$	−0.16	−0.73; 0.41	0.57
$\beta_{food}$	−0.47	−0.97; −0.02	0.06
$\beta_{time,\;Derby,\;animal}$	−0.05	−0.11; 0.01	0.1
$\beta_{time,\;Typhimurium,\;animal}$	−0.03	−0.11; 0.04	0.35
$\beta_{time,\;Derby,\;food}$	0.02	−0.02; 0.07	0.37
$\beta_{time,\;Typhimurium,food}$	−0.007	−0.07; 0.06	0.83
Ceftazidime:			
$\beta_{0}$	−0.6	−0.72; −0.52	-
$\beta_{Derby}$	−0.23	−0.72; 0.29	0.36
$\beta_{food}$	−0.05	−0.18; 0.32	0.38
$\beta_{time,\;Derby,\;animal}$	−0.03	−0.11; 0.04	0.4
$\beta_{time,\;Typhimurium,\;animal}$	−0.02	−0.05; 0.13	0.23
$\beta_{time,\;Derby,\;food}$	−0.04	−0.08; 0.001	0.06
$\beta_{time,\;Typhimurium,food}$	−0.0002	−0.03; 0.03	0.99
Cefotaxime:			
$\beta_{0}$	−2.7	−2.8; −2.6	-
$\beta_{Derby}$	0.53	0.2; 0.96	0.002
$\beta_{food}$	0.2	0.04; 0.35	0.017
$\beta_{time,\;Derby,\;animal}$	−0.02	−0.08; 0.053	0.49
$\beta_{time,\;Typhimurium,animal}$	0.03	0.011; 0.04	0.001
$\beta_{time,\;Derby,\;food}$	−0.06	−0.1; −0.02	0.003
$\beta_{time,\;Typhimurium,food}$	0.017	−0.001; 0.04	0.11

^1^$\beta_{0}$ stands for the intercept for serovar Typhimurium and matrix food. $\beta_{Derby}$ is the increment in the $\beta_{0}$ for the serovar Derby, and $\beta_{food}$ is the increment in the $\beta_{0}$ for the matrix food.$\;\beta_{time}$ coefficients are the log-linear change in the MIC per increase of year (slope) for the different combinations of serovar and matrix. ^2^ CI = Confidence interval.

**Table 4 pathogens-11-00905-t004:** Origin of *Salmonella* Derby and Typhimurium isolates over 16 years, belonging to culture collections of the Preventive Veterinary Medicine Department—Faculty of Veterinary Medicine of Federal University of Rio Grande do Sul (FAVET/UFRGS).

Collect Year	Number of Isolates—*Salmonella*		Reference
	Derby	Typhimurium	
2000 and 2001	25	37	Bessa et al., 2004 [50]
2001	14	30	Castagna et al., 2004a [51]
2001	9	29	Castagna et al., 2004b [52]
2001	0	7	Michael et al., 2002 [53]
2003	0	46	Silva et al., 2006 [54]
2005	28	25	Schwarz et al., 2011 [55]
2005 and 2006	9	8	Murmann et al., 2009 [30]
2008 and 2009	31	30	Silva et al., 2012 [14]
2010 and 2011	19	53	Pissetti et al., 2012 [13]
2012	0	5	Werlang et al., 2019 [56]
2014	0	2	Werlang et al., 2021 [57]
2015	0	6	Paim et al., 2019 [16]

## Data Availability

The data and scripts are available on the public repository: https://github.com/eduardodefreitascosta/AMR_surveillance.

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
