# Peer review of "Critically Important Antimicrobial Resistance Trends in Salmonella Derby and Salmonella Typhimurium Isolated from the Pork Production Chain in Brazil: A 16-Year Period"

_pathogens, 2022, doi:10.3390/pathogens11080905_

Round 1

Reviewer 1 Report

Antimicrobial resistence is a growing public health of broad concerne to countries in worldwide. The authors propose an interesting  study on antimicrobial resistence trends in Salmonella Derby and Salmonella Typhimurium isolated from pork production chain in Brazil. Nevertheless, I suggest, before publication some revision:

-          although the introduction considers several aspects of antibiotic resistance, with valid bibliographic references would be interesting a reference to the global burden associated with drug-resistant infections ARTICLES| VOLUME 399, ISSUE 10325, P629-655, FEBRUARY 12, 2022

-          ”Global burden of bacterial antimicrobial resistance in 2019: a systematic analysis. Antimicrobial Resistance Collaborators  Open AccessPublished:January 19, 2022DOI:https://doi.org/10.1016/S0140-6736(21)02724-0

-          Materials and methods:

-              Line 245 the authors indicated the time of strains  collection between 2000-2015 in the discussion is indicated as 2000-2016

-              Line 277: can Authors add description of positive control used for PCR

Reviewer 2 Report

Caroline Pissetti and colleagues (pathogens-1834477) represented an investigation of critically important antimicrobial resistance, as defined by WHO, in certain Salmonella serovars. The overall study is purely data mining and analysis, while the topics are of potential interest in general, however, the rational design and logic flow in the main text should be improved. Additionally, or more importantly, the origin data for the analysis is not available, this may be the key issue for further evaluation of this study.

1. Why do these two serovars matter, should this be well introduced and discussed? Why not study Salmonella as a whole?

2. Line 57-59. Your study was also purely descriptive? In terms of trend, why this issue is urgently needed. So, the real advancements you made in the field are not clear. 

3. Even though the trend investigation is studied, there are many studies that have already made this contribution (10.3389/fmicb.2021.702909; 10.3389/fmicb.2019.00985). The field has already been well documented, many new knowledge gaps should be adjusted, and this message should be clarified in a general picture not only in Brazil.  

4. The original data (in supplemental excel), with detailed meta knowledge, should be provided, this is the core part of this study. 

5. Some references are old and central in brazil, please introduce a big picture in the field, not just one single country with limited general interest to a global audience.

6. MIC assay, what are the drugs does? the diameter should change to the MIC value if possible, considering it has limited power for cross lab comparability.

7. mcr has ten variants, the remaining nine should also be scanned.

8. Statistical analysis is not clear to me, please clarify, it is hard to interpret for biologists.

9. Table 1, what is this "P-value" mean? and to whom it compares?

10. The beta value is not clear to me.

11. Considering a limited number of samplings with reused studied purposes, I suggest a brief report could be an ideal style for this work. 

Round 2

Reviewer 2 Report

none